# Virulence Profiling, Multidrug Resistance and Molecular Mechanisms of *Campylobacter* Strains from Chicken Carcasses in Tunisia

**DOI:** 10.3390/antibiotics11070830

**Published:** 2022-06-21

**Authors:** Awatef Béjaoui, Manel Gharbi, Sarra Bitri, Dorsaf Nasraoui, Wassim Ben Aziza, Kais Ghedira, Maryem Rfaik, Linda Marzougui, Abdeljelil Ghram, Abderrazek Maaroufi

**Affiliations:** 1Laboratory of Epidemiology and Veterinary Microbiology, Group of Bacteriology and Biotechnology Development, Institut Pasteur de Tunis, University of Tunis El Manar (UTM), Tunis 1002, Tunisia; manelgharbi2@gmail.com (M.G.); sarrabitri@gmail.com (S.B.); dorsafnasraoui74@gmail.com (D.N.); wassimbenaziza7@gmail.com (W.B.A.); maryemrfaik@gmail.com (M.R.); lindamarzougui@gmail.com (L.M.); abdeljelil.ghram@pasteur.tn (A.G.); abderrazak.maaroufi@pasteur.tn (A.M.); 2Laboratory of Bioinformatics, Biomathematics and Biostatistics, Institut Pasteur de Tunis, University of Tunis El Manar (UTM), Tunis 1002, Tunisia; ghedirakais@gmail.com

**Keywords:** *Campylobacter*, chicken, foodborne pathogens, resistance, virulence

## Abstract

Antibiotic resistance in foodborne pathogens is an emergent global health concern. The objectives of this study were to assess antimicrobial resistance (AMR) in *Campylobacter* isolates from chicken carcasses and to investigate the AMR molecular mechanisms as well as the presence of virulence determinants. The study was performed on 257 samples collected from abattoirs and retail shops in northeastern Tunisia. Forty-eight *Campylobacter* isolates were recovered and identified as *C. jejuni* (n = 33) and *C. coli* (n = 15). Antibiotic resistance was tested against eight antibiotics and high resistance rates were observed against tetracycline (100%), erythromycin (97.9%), ciprofloxacin (73%), nalidixic acid (85.4%), ampicillin (83.3%), amoxicillin/clavulanic acid (22.9%), chloramphenicol (75%), and gentamicin (27.1%). All isolates were multidrug-resistant, and 22 resistance patterns were found. All isolates were screened for AMR genes (*tet*(*O*), *tet*(*A*), *tet*(*B*), *tet*(*L*), *cmeB*, *ermB*, *bla_OXA-61_*, and *aphA-3*), and for point mutations in *gyrA* (C257T substitution) and *23SrRNA* (A2075G/A2074C) genes. All screened AMR genes, as well as the C257T and the A2075G mutations, were detected. The virulence genotypes were also determined, and all isolates carried the motility (*flaA*) and invasion (*cadF*) genes. Most of them also harbored the *cdtA*, *cdtB*, and *cdtC* genes, encoding the *Campylobacter* toxin. The screening of the *cgtB* and the *wlaN* genes, involved in Guillain-Barré Syndrome expression, revealed the presence of the *cgtB* in 21.2% of *C. jejuni* strains, whereas none of them carried the *wlaN* gene. Our findings highlight the emergence of *Campylobacter* strains simultaneously harboring several virulence and AMR determinants, which emphasizes the risk of transmission of MDR strains to humans via the food chain. Hence, controlling the dissemination of foodborne pathogens “from the farm to the fork” as well as restricting the use of antimicrobials in husbandry are mandatory to prevent the risk for consumers and to mitigate the dissemination of MDR pathogens.

## 1. Introduction

Thermotolerant *Campylobacter*, particularly *C. jejuni* and *C. coli*, are considered as the most common cause of bacterial foodborne illness in humans, both in developed and low-income countries [1]. *C*. *jejuni* is responsible for 80–90% of campylobacteriosis cases worldwide and may induce immunoreactive complications, such as polyarthralgia, Miller Fisher, and Guillain-Barré syndromes (GBS) [2,3]. According to the World Health Organization (WHO), *Campylobacter* is one of the four most important causes of diarrheal diseases leading to 550 million people falling ill yearly, including 220 million children under the age of five years [4]. The major source of human infection is the handling and consumption of contaminated chicken meat [5]. Indeed, poultry is the main reservoir of *Campylobacter*, and at the slaughter stage, broiler carcasses are easily contaminated after evisceration. Even carcasses from negative flocks can be contaminated by cross-contamination through slaughter processing [6]. Thus, the pathogen may be transported from the farm to the final product through the slaughtering process [7].

The pathogenesis of *Campylobacter* remains unclear. However, specific pathogenic properties appear to contribute to the bacterial survival and establishment of disease in the host. Indeed, many virulence factors, including motility, adhesion, colonization, host cell invasion, and toxin production, are associated with *Campylobacter* strain virulence [8]. The carbohydrate structure (lipooligosaccharides) was shown to be responsible for triggering an excessive immune response, which provokes severe complications, such as GBS [8]. Therefore, the identification of genes associated with virulence in *Campylobacter* strains provides a better understanding of their pathogenicity and their ability to cause disease.

The treatment of campylobacteriosis is based mainly on the use of quinolones (e.g., ciprofloxacin), macrolides (e.g., erythromycin), and tetracyclines [6]. In the last decade, several studies have reported the emergence of AMR in *Campylobacter* and the transmission of MDR strains to humans, predominantly through food consumption [9]. Absolutely, rising prevalences of AMR among Gram-negative bacteria have become an emerging global threat [10]. Indeed, the overuse of antibiotics in human medicine and husbandry, particularly in developing countries, has led to the emergence of antibiotic resistance in bacteria and the spread of MDR strains [10], which increases the risk of even banal infections becoming fatal. In Tunisia, the absence of a national surveillance system for antimicrobial use in veterinary fields combined with the free sale of drugs constitute one of the main reasons behind the steady increase in antimicrobial resistance in husbandry [11,12].

Furthermore, the emergence of MDR strains harboring several virulence determinants simultaneously might have an additive influence on the severity of infections. Therefore, the surveillance of foodborne pathogen dissemination, AMR prevalence, and virulence factor characterization are critical pillars in epidemiological investigations, risk management, and control strategies established in poultry farming and meat chain production.

Keeping in view the current situation, the present study aimed to assess the contamination rate with *Campylobacter* in chicken carcasses, determine the AMR prevalence, profile MDR patterns, and characterize genes conferring AMR and virulence factors.

## 2. Results

### 2.1. Contamination Rates of Campylobacter spp. in Chicken Carcasses

Based on the culture method and the biochemical tests, a total of 56 presumptive *Campylobacter* isolates were recovered. Isolates showed on Karmali agar greyish, flat, and moistened colonies, with a metallic sheen and regular edge. Microscopic examination showed cells with characteristic corkscrew-like motility. The isolates were oxidase/catalase positive and Gram-negative curved bacilli.

Out of the 56 presumptive *Campylobacter* isolates, 48 were confirmed by PCR genus identification. Therefore, the overall isolation rate (IR) was 18.7% (n = 48/257), including 68.7% (n = 33/48) of *C. jejuni* and 31.2% (n = 15/48) of *C. coli*. The IR of *Campylobacter* in samples from abattoirs was 22.5% (n = 32/142) with 68.7% (22/32) of *C. jejuni* as the predominant species and 31.2% (10/32) of *C. coli*. Meanwhile, the IR of *Campylobacter* from retail shops samples was 13.9% (n = 16/115), with 68.7% (11/16) of isolates as *C. jejuni* and 31.2% (5/16) as *C. coli* (Table 1).

### 2.2. Antimicrobial Resistance: Phenotypic Patterns

The phenotypic antimicrobial susceptibility testing to eight antimicrobial agents revealed high resistance rates. Indeed, all isolates were resistant to tetracycline, and only one strain was susceptible to erythromycin. The resistance to ampicillin was observed in 83.3%, while isolates seemed more sensitive to amoxicillin/clavulanic and 77.1% were susceptible. Resistance rates to ciprofloxacin, nalidixic acid, and chloramphenicol were 73%, 85.4%, and 75%, respectively. Thirteen strains (27.1%) were resistant to gentamicin. The distribution of antimicrobial resistance rates among *C. jejuni* and *C. coli* species from abattoirs and retail stores are presented in Table 2.

Multidrug resistance to at least three antimicrobial classes was detected in all *Campylobacter* isolates, and the rates of resistance profiles, including three, four, five, and six antimicrobial classes, were 4.7%, 33.3%, 41.7%, and 20.8%, respectively. Most of *C. jejuni* (33.3%) and *C. coli* (46.6%) isolates exhibited different MDR patterns against five antimicrobial classes. Lower percentages of isolates were resistant to six classes (24.2% of *C. jejuni* vs. 13.3% of *C. coli*). Overall, 22 resistance patterns were detected (Table 2), and the most common pattern, “ERY AMP CIP NAL CHL TET”, was present in 27.1% of *Campylobacter* isolates. The percentage of *C. coli* strains (five out of 15 equivalent to 33.3%) belonging to this group is higher than the percentage of *C. jejuni* isolates (eight out of 33 equivalent to 24.2%). The second most prevalent pattern, “ERY AMP CIP NAL TET”, was detected in 10.4% (five out of 48) of isolates. The remaining patterns comprised less than 10% of isolates. For *C. coli*, 10 profiles composed of 3–6 antimicrobial classes were detected. While for *C. jejuni*, 18 patterns composed of three (one pattern), four (eight patterns), five (six patterns), and six (four patterns) antimicrobial classes were detected. Seven patterns were detected in both species (Table 3).

### 2.3. Detection of Antimicrobial Resistance Genes

The selected AMR genes were evaluated in all phenotypically *Campylobacter* resistant isolates. When screened for tetracycline resistance genes, all strains (n = 48) harbored the *tet*(*O*) gene. The *tet*(*A*) was also prevalent at 89.6% (43/48), whereas *tet*(*B*) was detected in 10 isolates (20.8%) and *tet*(*L*) only in two strains (4.1%). The *tet* genes’ identity, except the *tet*(*L*) gene, was confirmed by sequencing. The obtained sequences, compared with the corresponding public sequences of antibiotic-resistant bacteria available in the GenBank database, showed a nucleotide similarity of 99–100%. Sequencing failed for the *tet*(*L*) amplicons, likely because both positive isolates showed a weak amplification signal.

The resistance-associated point mutation (C257T) in the quinolone resistance determining regions (QRDR) of *gyrA* gene (encoding the subunit A of DNA gyrase) was detected in 81.8% (36/44) of resistant strains. All the strains with a CIP+NAL resistant phenotype (33/33) and two ciprofloxacin resistant strains (2/3) had this mutation, whereas, for the nalidixic acid resistant isolates (n = 8), only one strain harbored the C257T mutation.

The A2075G mutation in the *23S rRNA* gene conferring resistance to macrolides was observed in 76.6% (36/47). Nine erythromycin-resistant isolates (19.1%) harbored the A2074C mutation, whereas 48.9% (23/47) carried the *ermB* gene (encoding a ribosomal methylase). Meanwhile, none of these macrolide resistance-associated determinants was detected in four isolates (8.5%).

The presence of the β-lactamase *bla_OXA-61_* gene was observed in 21.4% (9/42) of the β-lactams resistant strains, while 46.1% (6/13) of isolates carried the *aphA-3* gene (encoding an aminoglycoside 3′-phosphotransferase) conferring resistance to aminoglycosides. Interestingly, all isolates carried the *cmeB* gene encoding the subunit B of the cmeABC pump efflux.

### 2.4. Prevalence of Virulence Determinants in Campylobacter Isolates

In general, virulence gene prevalence in *Campylobacter* strains was high, and strains carried an average of six virulence genes. The most frequently identified genes were as follows: *cadF* (encoding a 37-kDa outer membrane protein binding to host fibronectin), *flaA* (coding for flagellin), *cdtA*, *cdtB*, and *cdtC* (encoding the cytolethal-distending toxin). Indeed, all the strains (100%) were positive for the *flaA* and *cadF* genes. Cytotoxin production genes were also highly prevalent, particularly in *C. coli*. Indeed, all isolates were positive for the three *cdt* genes. The *cdtA*, *cdtB*, and *cdtC* genes were also prevalent in *C. jejuni* isolates, with the percentage of 88% (n = 29/33), 72.7% (n = 24/33), and 75.7% (n = 25/33), respectively. The *ceuE* gene (encoding the enterochelin uptake substrate-binding protein, involved in iron acquisition) was carried by 93.9% (n = 31/33) of *C. jejuni* isolates and 33.3% (n = 5/15) of *C. coli*. *virB11*, involved mainly in cell invasion, was detected in 36.4% of *C. jejuni* (n = 12/33) and 40% of *C. coli* (n = 6/15). The *cgtB* and *wlaN* genes, both encoding a β-1, 3-galactosyltransferase enzyme and involved in triggering Guillain-Barré syndrome (polyneuropathic disorder), were also screened. The *cgtB* gene was detected in 21.2% (n = 7/33) of *C. jejuni*, while none of the isolates harbored the *wlaN* gene. When looking at the virulence patterns, 10 virulotypes were observed (Table 4). The most prevalent was the combination “*flaA, cadF, cdtA, cdtB, cdtC, ceuE*” detected in 13 isolates, followed by the combinations “*flaA, cadF, cdtA, cdtB, cdtC*”, “*flaA, cadF, cdtA, cdtB, cdtC, ceuE, virB11*”, and “*flaA, cadF, cdtA, cdtB, cdtC, ceuE, virB11, cgtB*” detected in eight, seven, and six strains. Moreover, it was observed that *C. jejuni* strains presented a higher percentage of virulence genes (63.7% harbored six genes or more) than *C. coli* (only 46.7% carried six genes), where *p* < 0.05.

## 3. Discussion

Human campylobacteriosis is mainly caused by consuming contaminated foods from poultry origin. In the current study, we assessed the *Campylobacter* contamination rate in chicken carcasses. The isolation rate of *Campylobacter* was 18.7%, which was within the range of reported rates in different studies, such as in Italy (17.38%) [13]. By contrast, higher rates were recorded in China (77%) [14] and South Korea (54.1%) [7]. In our previous study on *Campylobacter* prevalence in broiler flocks [15], we reported contamination rates ranging from 6.1% to 56%. Hence, the contamination rate at the slaughter level was in line with these results. The total isolation rate of *Campylobacter* from samples originating from slaughterhouses (23.2%) was higher than retail shops (13%), which might be attributed to the effect of refrigeration and frozen conditions on the viability of *Campylobacter* on carcasses. Indeed, it was reported that refrigeration and freezing exert a lethal effect on *Campylobacter* cells [16,17]. In the present study, 68.7% of *Campylobacter* isolates were identified as *C. jejuni* and 31.2% as *C. coli*. The predominant species recovered from poultry carcasses was *C. jejuni*, as reported in several studies worldwide [18]. Indeed, it was shown in multiple studies that the most recovered thermotolerant species at the end of the poultry meat chain production is *C. jejuni* [19,20].

Antimicrobial resistance is an emerging public health issue, compromising antibiotic treatment of human and animal infections. The development of antimicrobial resistance in foodborne pathogens is a result, in part, of the misuse of veterinary drugs in husbandry. In intensive poultry production, antibiotics have been commonly used whether in bacterial infection control or for growth promotion [21,22]. The emergence and dissemination of multi-drug resistant *Campylobacter* strains from different origins were described worldwide, and resistant strains might be transmitted to humans via the food chain [23]. Even though the use of antibiotics as growth promoters is regulated in Tunisia, the overuse of antibiotics in husbandry remains a real concern because open access to antibiotics from local vendors is usually possible. In this study, high resistance rates were obtained for almost all tested antibiotics except amoxicillin/clavulanic acid and gentamicin, which is in total agreement with our previous results from broiler flocks [15]. This underlines the importance of limiting the use of antibiotics at the farm level to reduce the emergence of drug-resistant strains and their spread in food animal products. Regarding macrolides, the frontline antibiotics for *Campylobacter* infection treatment, most isolates (97% in *C. jejuni* vs. 100% in *C. coli*) were resistant to erythromycin, which is consistent with many reports elsewhere [24,25]. The increase of macrolide resistance rates is likely attributable to the widespread use of macrolides, such as tylosin, in poultry production. Resistance to erythromycin is mainly caused by mutations at positions A2075G and/or A2074C of the domain V of the *23S rRNA* gene [26]. The presence of the *ermB* gene encoding the 23S rRNA methyltransferase and the cmeABC multidrug efflux pump has also been shown to be involved in the acquired resistance to erythromycin [27]. In this study, the A2075G mutation was detected in 76.6% of isolates, the A2074C in 19.1%, and the *ermB* gene in 48.9% of isolates. Four out the 47 erythromycin-resistant strains did not carry any of the selected macrolide resistance determinants. However, all of these isolates carried the *cmeB* gene which can explain this resistance due to the cmeABC efflux pump expression, as suggested by other studies [26].

In the present study, we revealed high resistance rates to ciprofloxacin (60% in *C. jejuni* vs. 86.7% in *C. coli*) and nalidixic acid (87.9% in *C. jejuni* vs. 80% in *C. coli*). Such results are consistent with reports from Algeria, Poland, and Belgium [28,29,30]. Quinolone-resistance has been linked with the presence of two different mechanisms, namely the presence of the *cmeABC* operon and point mutations in the quinolone-resistance determining region (QRDR) of the DNA gyrase A (*gyrA*) gene. In addition to several other mutations in *gyrA* (Thr86Lys, Thr86Ala, Ala87Pro, Asp90Tyr, Asp90His, etc.) [31], various mutations in *gyrB* might be involved in quinolones and FQ-resistance. *Campylobacter* isolates resistant to ciprofloxacin and/or nalidixic acid were screened for the presence of the C257T mutation in the *gyrA* gene. The results showed that this mutation was not detected in some resistant isolates, particularly those resistant only to nalidixic acid. This observation is in agreement with previous reports suggesting that this point mutation does not confer universal resistance to all quinolone antibiotics and arguing that quinolone-resistance might be associated with other unknown resistance mechanisms [30].

Given the long-term use of tetracyclines in food animal production and poultry raising, large numbers of tetracycline-resistant isolates were found in animal reservoirs. It has been previously reported that 95.6% of *C. jejuni* and 97.5% of *C. coli* isolates from chicken in China were resistant to tetracycline [32]. In the present study, 100% of isolates were resistant to tetracycline. Besides, the genetic analysis was fully concordant. Indeed, all strains were shown to carry the gene *tet*(*O*), which is known to be responsible for tetracycline resistance in *Campylobacter* strains [33]. We have also shown the presence of *tet*(*A*) in most isolates (89.6%), which corroborates the results of Nguyen et al. [34] that reported the detection of *tet*(*A*) gene in 90.3% of *C. jejuni* and 100% *C. coli* isolates from chicken in Kenya, in contrast to the results from china (6.5%) and Iran (18%) [25,35]. At a lesser level, we detected the *tet*(*B*) in 20.8% of isolates, and the *tet*(*L*) variant in only two strains (4.1%). A previous study in Tunisia has shown that nine *tet* genes (*tet*(*A*), *tet*(*B*), *tet*(*K*), *tet*(*L*), *tet*(M), *tet*(O), *tet*(Q), *tet*(S), and *tet*(X)) were found in 100% of analyzed gut broiler chickens lots [36]. The same study reported that the *tet*(*O*) was present in 98% of samples, the *tet*(*A*) in 90.2%, and the *tet*(*B*) in 76.4%. In contrast to our results, they showed that *tet*(*L*) was also prevalent and it was detected in 98% of samples. This gene encoding an efflux pump, often reported in methicillin-resistant *Staphylococcus aureus* (MRSA) and conferring resistance to tetracycline, has been rarely described in *Campylobacter* [37]. The *tet*(*L*) was confirmed to confer an increased resistance to tigecycline, which is considered as the last antibiotic choice to treat infections caused by carbapenem-resistant *Enterobacteriaceae*. *tet*(*A*) also conferred increased resistance to this antibiotic, which suggests that the tetracycline efflux pump, whether encoded by *tet*(*A*) or *tet*(*L*), can contribute to increased tigecycline resistance. Besides, comparative genomic analysis indicated that the *tet*(*L*) variant was located within a multidrug resistance genomic island (MDRGI) in the *Campylobacter* spp. chromosome. In this MDRGI, a florfenicol resistance gene, *fexA*, and a tetracycline resistance gene, *tet*(*O*), were present, which suggests that the usage of tetracyclines and florfenicol in husbandry could increase the prevalence and the dissemination of the *tet*(*L*) variant by selection pressure [37].

The resistance level against ampicillin (β-lactam) (88% in *C. jejuni* vs. 73.3% in *C. coli*) was important, whereas the combination amoxicillin/clavulanic acid seemed to remain effective against *Campylobacter* (27.3% in *C. jejuni* vs. 13.3% in *C. coli*), which is consistent with our previous results [15], and with other reports, e.g., from Australia [38]. Even though resistance to β-lactams has been widely reported among *Campylobacter*, the mechanism of resistance to ampicillin and the involvement of β-lactamase genes are not well understood. The acquisition of the *bla_OXA-61_* gene seems to be associated with this resistance. However, *C. jejuni* can produce more than one type of β-lactamase. Indeed, other uncharacterized beta-lactamase genes were reported in *Campylobacter* [39]. The CmeABC also plays an important role in the resistance to beta-lactam drugs [26].

Resistance towards gentamicin is a novel phenomenon in *Campylobacter* isolates as it is used for treating systemic infections, and low resistance proportions have been widely documented [40]. However, recent studies reported increased resistance rates against this antibiotic [41], which corroborated with our findings. In fact, we report that 30.3% of *C. jejuni* and 20% of *C. coli* isolates are resistant to gentamicin. When screened for the *aphA-3* gene, 46.1% of resistant isolates were positive, and this result is in line with previous reports [42]. Other genes, such as *aacA4* and *aphA-7*, commonly associated with aminoglycosides resistance should be investigated to better characterize resistance to aminoglycosides for these strains.

In Tunisia, as in many countries worldwide, the use of chloramphenicol is prohibited in husbandry [43]. Nevertheless, we noted high levels of resistance towards this drug (72.7% in *C. jejuni* vs. 80% in *C. coli*), which is similar to other reports, e.g., from India [44]. These results could be explained by the frequent use of florfenicol as a broad-spectrum antibiotic in veterinary medicine, resulting in a combined acquired resistance to florfenicol/chloramphenicol [45].

Despite the presence of small resistance discrepancies between phenotypes and genotypes in some strains, the presence of *cmeABC* pump efflux in all strains might be responsible for the observed phenotypic resistance [46]. Indeed, as we have shown in our previous study [47], the presence of the *cmeABC* pump efflux might be the most potential mechanism of resistance in strains lacking AMR genes. However, more in-depth investigation should be performed to determine other putative AMR molecular mechanisms.

Multidrug resistance was observed in all isolates, which was not unexpected due to the high MDR rates previously observed in *Campylobacter* strains from broiler flocks in Tunisia [15]. Twenty-two resistance patterns were found, and several of them were detected in poultry flocks. Resistance to five and six groups of antimicrobials was observed in 62.5% of strains. In addition, 91% of isolates were resistant to erythromycin, quinolones, and tetracycline.

Thus, these results highlighted that the health risk to the consumer is further complicated by limiting treatment options. Therefore, the clinical treatment of campylobacteriosis and probably other enterobacterial infections should be carefully reconsidered. Moreover, the detection of all screened AMR determinants suggests that this zoonotic agent would be a potential reservoir for the dissemination of resistance determinants to several intestinal pathogens by horizontal gene transfer.

In the second part of this study, we analyzed the presence of virulence factors in both *C. jejuni* and *C. coli* and observed that most of the screened virulence genes were prevalent in both species. The results showed that the most common were *flaA* and *cadF*, which have been associated with the bacterial adherence capacity to epithelial cells [8]. This result corroborated with several previous studies [48,49,50]. The high prevalence of these genes among *Campylobacter* strains can be explained by the fact that they are the key to the gut colonization process [49,50,51]. The *cdtA*, *cdtB*, and *cdtC* genes, which cause diarrhea by interfering with the division and differentiation of cells in the intestinal crypt, were detected in all *C. coli* strains and most of *C. jejuni* strains. These results are in line with other studies, which have described the high prevalence of these genes in *Campylobacter* strains [52]. The *ceuE* gene involved in gut colonization [53] was more likely to occur among *C. jejuni* (93.9%) than in *C. coli* (33.3%) strains. Previous studies described the high prevalence of this gene in different *C. jejuni* isolates and its importance in virulent *C. jejuni* strains, both for human and poultry [54]. The presence of *wlaN* and *cgtB* genes is correlated with a higher ability of strain invasiveness, since they are encoding for a β-1,3-galactosyltransferase enzyme associated with the production of sialylated lipooligosaccharide, which is an important factor in triggering the Guillain-Barré and Miller-Fisher syndromes in patients after *C. jejuni* infection. In this study, 21.2% of strains carried the *cgtB* gene. However, none of them harbored the *wlaN* gene. While several studies reported the coexistence of *cgtB* and *wlaN* in strains, Guirado [55] described a differential prevalence of the genes *wlaN* and *cgtB* in *C. jejuni* isolates, which corroborated with our findings.

Overall, *C. jejuni* strains contained eight virulence gene patterns, with an average of six virulent factors. On the other hand, *C. coli* presented three different virulence gene patterns with the presence of at least five virulent determinants, including *FlaA*, *cadF*, *cdtA*, *cdtB*, and *cdtC*. Previous studies have shown that *C. jejuni* harbored more virulence-associated genes than *C. coli*, which might contribute to the survival and colonization of *C. jejuni* in the poultry gut, as well as to the development of the disease in humans [56].

In terms of this study, the results underscore the need for “one-health” approaches to control the dissemination of foodborne pathogens and to mitigate AMR emergence and the dissemination of virulent strains.

## 4. Materials and Methods

### 4.1. Sample Collection

A total of 257 chicken carcass samples were collected between 2017 and 2018, from two large poultry slaughterhouses (n = 142) located in northeastern Tunisia and 17 local randomly selected retail stores (n = 115) from different grocery chains in the district of Grand Tunis. Samples belonging to different brands of conventional chickens were aseptically collected in separate sterile plastic bags and were immediately transported in a cool box. Samples processing was performed in the same day.

### 4.2. Isolation of Campylobacter and Identification

Isolation of *Campylobacter* spp. from carcasses was achieved in accordance with the ISO 10272-1:2017(E) method, with slight modifications. Briefly, each sample (25 g) was subjected to an enrichment step in 225 mL of Bolton broth (Oxoid, Basingstoke, UK), supplemented with the Bolton broth selective cocktail (Oxoid, Basingstoke, UK), and incubated at 42 °C for 24 h under microaerophilic conditions (5% O_2_, 10% CO_2_, and 85% N_2_) using GENbox generators (BioMerieux, Craponne, France). From the enrichment culture, the selective Karmali agar plates (SIGMA-ALDRICH, Bangalore, India) were inoculated. The plates were incubated for 48 h under the same conditions, as above. The suspected *Campylobacter* colonies were examined for cell morphology and motility under optic microscope and then inoculated on non-selective blood agar for oxidase/catalase reactions and Gram coloration.

For molecular identification, bacterial DNA was extracted by the boiling method as described previously [15]. The genus confirmation was performed using a specific PCR amplification of the *16S rRNA* gene, as described by Linton [57]. Species identification of the confirmed *Campylobacter* isolates as *C. jejuni* or *C. Coli* was based on *mapA* and *ceuE* gene amplification [58] using the primer sets listed in Appendix A. The reference strains *C. jejuni* (ATCC 33291) and *C. coli* (CCUG 11283-T) were used as positive controls.

### 4.3. Antimicrobial Susceptibility Testing

Antimicrobial susceptibility testing (AST) was performed using the Kirby-Bauer disk diffusion method, in compliance with the European Committee on Antimicrobial Susceptibility Testing guidelines [59]. In brief, bacterial suspensions were adjusted to 0.5 McFarland turbidity standard and then plated onto Mueller-Hinton agar (Bio life, Milan, Italy) supplemented with 5% defibrinated horse blood. Afterward, the antimicrobial disks were distributed on the plates and incubated at 37 °C for 24 h. The tested antimicrobials were: erythromycin (ERY: 15 µg), ciprofloxacin (CIP: 5 μg), nalidixic acid (NAL: 30 μg), tetracycline (TET: 30 μg), amoxicillin/clavulanic acid (AMC: 20/10 μg), ampicillin (AMP: 10 μg), gentamicin (GEN: 10 μg), and chloramphenicol (CHL: 30 μg). Inhibition zones were interpreted according to the EUCAST (2017) breakpoints for *Campylobacter* [59]. The inhibition zone diameters interpreted as intermediate were considered as resistant. Isolates resistant to at least three antimicrobial classes were defined as multi-drug resistant (MDR).

### 4.4. Molecular Detection of Antimicrobial Resistance and Virulence Genes

Isolates resistant to the tested antibiotics were screened for the presence of the correspondent AMR genes. The *tet*(*O*), *tet*(*A*), *tet*(*B*), and *tet*(*L*) (tetracycline resistance) were amplified by PCR. The amplicons of *tet* genes were purified using the QIAquick Gel extraction Kit (QIAGEN, Hilden, Germany), and both DNA strands were sequenced. The obtained sequences were aligned by Blast (https://blast.ncbi.nlm.nih.gov/Blast.cgi, accessed on 21 January 2022), with the public sequences available in GenBank.

The macrolid resistance mechanism was investigated by targeting the A2074C and A2075G mutations in the23S rRNA gene, using the Mismatch Amplification Mutation Assay (MAMA-PCR), as described by Alonso [60], and by detecting the *ermB* gene by PCR. The resistance to quinolones was examined by MAMA-PCR, as described by Zirnstein [61,62], to detect the C257T (Thr-86-Ile) mutation in the *gyrA* gene in *C. jejuni* (ACA-ATA) and *C. coli* (ACT-ATT). The genetic determinants conferring resistance toward β-lactams and aminoglycosides were investigated by targeting the *bla_OXA-61_* and *aphA-3* genes, respectively. Besides, the cmeABC multidrug efflux pump conferring unspecific resistance to all the tested drug families was investigated by the amplification of the *cmeB* gene.

Regarding the virulence patterns assessment, *Campylobacter* isolates were screened by PCR for nine putative virulence-associated genes. The main virulence factors involved in motility (*flaA*), adhesion (*cadF*), invasion (*virB11*), production of the only *Campylobacter* toxin, cytolethal-distending toxin (*cdtA*, *cdtB*, and *cdtC*), colonization (*ceuE*: encoding a protein-binding-dependent transport system for the siderophore enterochelin), and in the expression of severe complications, such as the Guillain-Barré and Müller Fisher syndromes (*cgtB*, and *wlaN*) were examined.

All the PCR primer sets used in this study are shown in Appendix A.

### 4.5. Statistical Analysis

All data collected in this study were analyzed using R language. Data, including isolation rate, antibiotic resistance (susceptibility and resistance), and molecular detection of genes (positive or negative), were analyzed as absolute values and reported as percentages.

## 5. Conclusions

Although the sample size and the number of *Campylobacter* strains were limited, this study revealed high resistance rates against most of the tested antimicrobials, and all strains exhibited MDR profiles, suggesting a potential hazard to consumers. All screened AMR molecular mechanisms were detected, and most strains co-harbored several virulence-associated markers, which might increase bacterial pathogenicity and treatment failure. Hence, the current data highlight the need to strengthen control strategies of foodborne pathogens in husbandry as a potential reservoir of AMR spread and to implement an urgent national plan for curbing the AMR trend, in animals and humans, in a “one health” approach.

## Figures and Tables

**Table 1 antibiotics-11-00830-t001:** Isolation rates of thermotolerant *Campylobacter* strains from chicken carcasses.

Sources	No. of Samples	No. of *Campylobacter* Isolates (%)		
		*C. jejuni*	*C. coli*	Total
Abattoirs	142	22 (68.7%)	10 (31.2%)	32 (22.5%)
Stores	115	11 (68.7%)	5 (31.2%)	16 (13.9%)
Total	257	33 (68.7%)	15 (31.2%)	48 (18.7%)

**Table 2 antibiotics-11-00830-t002:** Antimicrobial resistance rates in *C. jejuni* and *C. coli* strains.

Sources	Species	No.	Antimicrobial Resistance Rates (n*)							
			ERY	AMP	AMC	CIP	NAL	CHL	TET	GEN
**Abattoirs**	*C. jejuni*	22	100% (22)	81.8% (18)	22.7% (5)	86.4% (19)	94.4% (21)	72.7% (16)	100% (22)	27.3% (6)
	*C. coli*	10	100% (10)	60% (6)	10% (1)	100% (10)	90% (9)	80% (8)	100% (10)	30% (3)
**Stores**	*C. jejuni*	11	90.9% (10)	100% (11)	36.4% (4)	36.4% (4)	72.7% (8)	72.7% (8)	100% (11)	36.4% (4)
	*C. coli*	5	100% (5)	100% (5)	20% (1)	60% (3)	60% (3)	80% (4)	100% (5)	-
**Total**		48	97.9% (47)	83.3% (40)	22.9% (11)	73% (35)	85.4% (41)	75% (36)	100% (48)	27.1 (13)

n*: number of resistant strains. ERY, erythromycin; AMP, ampicillin; AMC, amoxicillin/clavulanic acid; CIP, ciprofloxacin; NAL, nalidixic acid; CHL, chloramphenicol; TET, tetracycline; GEN, gentamicin.

**Table 3 antibiotics-11-00830-t003:** MDR patterns of *C. jejuni* and *C. coli* isolates from chicken carcasses.

ATB Profiles	Antimicrobial Groups	Isolates (n = 48)	Cj (n = 33)	Cc (n = 15)
ERY AMP AMC CIP NAL CHL TET GEN	*6*	3 (6.25%)	*3*	-
ERY AMP AMC NAL CHL TET GEN	6	1	1	-
ERY AMP CIP NAL CHL TET GEN	6	4 (8.3%)	2	2
ERY AMP NAL CHL TET GEN	6	1	1	-
AMP CIP NAL CHL TET GEN	5	1	1	-
ERY AMP CIP NAL CHL TET	5	13 (27.1%)	8	5
ERY AMP NAL CHL TET	5	3 (6.25%)	2	1
ERY AMP AMC NAL CHL TET	5	1	1	-
ERY AMP AMC CIP NAL CHL TET	5	1	1	-
ERY CIP NAL CHL TET GEN	5	1	-	1
ERY AMP AMC CHL TET GEN	5	1	1	-
ERY AMP NAL TET	4	1	1	-
ERY AMP CHL TET	4	2	1	1
ERY AMP CIP NAL TET	4	5 (10.4%)	4	1
ERY AMP AMC CIP NAL TET	4	1	1	-
ERY NAL CHL TET	4	1	1	-
ERY AMC CIP NAL TET	4	2	1	1
ERY CIP CHL TET	4	2	1	1
ERY CIP NAL TET GEN	4	1	1	-
ERY AMP AMC CIP TET	4	1	-	1
ERY AMP TET	3	1	1	-
ERY CIP NAL TET	3	1	-	1

Abbreviations: ERY, erythromycin; CIP, ciprofloxacin; NAL, nalidixic acid; TET, tetracycline; AMP, ampicillin; AMC, amoxicillin/clavulanic acid; CHL, chloramphenicol; GEN, gentamicin.

**Table 4 antibiotics-11-00830-t004:** AMR determinants and virulence profiles of *Campylobacter* strains.

	Strain	AMR Genotypes	Virolotypes
1	*C. jejuni*	*cmeB, ermB, Ery75, tet(O), tet(A), C257T*	*flaA, cadF, cdtA, cdtB, cdtC, ceuE, virB11, cgtB*
2	*C. jejuni*	*cmeB, Ery75, tet(O), tet(A), C257T*	*flaA, cadF, cdtA, cdtB, cdtC, ceuE, virB11, cgtB*
3	*C. jejuni*	*cmeB, Ery75, tet(O), tet(A), C257T*	*flaA, cadF, cdtA, cdtB, cdtC, ceuE, virB11, cgtB*
4	*C. jejuni*	*cmeB, Ery74, tet(O), tet(A), C257T*	*flaA, cadF, cdtA, cdtB, cdtC, ceuE, virB11, cgtB*
5	*C. jejuni*	*cmeB, tet(O), tet(B), C257T, bla_OXA-61_, aphA-3*	*flaA, cadF, cdtA, cdtB, cdtC, ceuE, virB11, cgtB*
6	*C. jejuni*	*cmeB, ermB, Ery75, tet(O), tet(A), bla_OXA-61_*	*flaA, cadF, cdtA, cdtB, cdtC, ceuE*
7	*C. jejuni*	*cmeB, ermB, ery75, Ery74, tet(O), tet(A), tet(B), tet(L), C257T*	*flaA, cadF, cdtA, cdtB, cdtC, ceuE, virB11*
8	*C. jejuni*	*cmeB, Ery75, tet(O), tet(A), C257T, aphA-3*	*flaA, cadF, cdtA, cdtB, cdtC, ceuE*
9	*C. jejuni*	*cmeB, ermB, Ery75, Ery74, tet(O), tet(A), C257T*	*flaA, cadF, cdtA, cdtB, ceuE, cgtB*
10	*C. jejuni*	*cmeB, Ery75, tet(O), tet(A), C257T*	*flaA, cadF, cdtA, ceuE*
11	*C. jejuni*	*cmeB, Ery75, tet(O), tet(A), C257T*	*flaA, cadF, ceuE, virB11*
12	*C. jejuni*	*cmeB, ermB, Ery75, tet(O), tet(A), tet(B), tet(L), C257T, bla_OXA-61_*	*flaA, cadF, cdtA, cdtB, cdtC, ceuE, virB11*
13	*C. jejuni*	*cmeB, Ery75, Ery74, tet(O), tet(A), C257T*	*flaA, cadF, ceuE*
14	*C. jejuni*	*cmeB, Ery75, tet(O)*	*flaA, cadF, ceuE*
15	*C. jejuni*	*cmeB, ermB, Ery75, tet(O), tet(A), C257T*	*flaA, cadF, cdtA, ceuE*
16	*C. jejuni*	*cmeB, ermB, Ery75, tet(O), tet(A), tet(B), C257T*	*flaA, cadF, cdtA, cdtB, cdtC*
17	*C. jejuni*	*cmeB, Ery75, tet(O), tet(A), C257T*	*flaA, cadF, cdtA, cdtC*
18	*C. jejuni*	*cmeB, ermB, Ery75, tet(O), tet(A), C257T*	*flaA, cadF, cdtA, cdtB, cdtC, ceuE*
19	*C. jejuni*	*cmeB, Ery75, tet(O), tet(A), C257T, bla_OXA-61_*	*flaA, cadF, cdtA, cdtB, cdtC, ceuE*
20	*C. jejuni*	*cmeB, ermB, Ery75, tet(O), tet(A), C257T*	*flaA, cadF, cdtA, ceuE*
21	*C. jejuni*	*cmeB, ermB, Ery75, tet(O), tet(A), C257T*	*flaA, cadF, cdtA, cdtB, cdtC, ceuE, virB11, cgtB*
22	*C. jejuni*	*cmeB, Ery74, tet(O), aphA-3*	*flaA, cadF, cdtA, cdtB, cdtC, ceuE*
23	*C. jejuni*	*cmeB, Ery75, tet(O), tet(A), tet(B), C257T, bla_OXA-61_*	*flaA, cadF, cdtA, cdtB, cdtC, ceuE*
24	*C. jejuni*	*CmeB, ermB, Ery75, tet(O), tet(A), tet(B)*	*flaA, cadF, ceuE*
25	*C. jejuni*	*cmeB, ermB, Ery75, tet(O), tet(A)*	*flaA, cadF, cdtA, cdtB, cdtC, ceuE, virB11*
26	*C. jejuni*	*cmeB, tet(O), tet(A), C257T*	*flaA, cadF, cdtA, cdtB, cdtC, ceuE*
27	*C. jejuni*	*cmeB, ermB, Ery74, tet(O), tet(A), tet(B), C257T, bla_OXA-61_*	*flaA, cadF, cdtA, cdtB, cdtC, ceuE, virB11*
28	*C. jejuni*	*cmeB, Ery75, tet(O), tet(A)*	*flaA, cadF, cdtA, cdtB, cdtC, ceuE, virB11*
29	*C. jejuni*	*cmeB, Ery75, Ery74, tet(O), tet(A)*	*flaA, cadF, cdtA, cdtB, cdtC, ceuE virB11*
30	*C. jejuni*	*cmeB, tet(O), bla_OXA-61_, aphA-3*	*flaA, cadF, cdtA, cdtB, cdtC, ceuE*
31	*C. jejuni*	*cmeB, Ery75, tet(O), tet(A), tet(B)*	*flaA, cadF, cdtA, cdtC, ceuE, virB11*
32	*C. jejuni*	*cmeB, ermB, tet(O), tet(A), C257T*	*flaA, cadF, cdtA, cdtB, cdtC, ceuE, virB11*
33	*C. jejuni*	*cmeB, tet(O), tet(A), C257T, bla_OXA-61_*	*flaA, cadF, cdtA, cdtB, cdtC, ceuE, virB11*
34	*C. coli*	*cmeB, Ery75, tet(O), tet(A), C257T, aphA-3*	*flaA, cadF, cdtA, cdtB, cdtC, ceuE*
35	*C. coli*	*cmeB, ermB, tet(O), tet(A), C257T, aphA-3*	*flaA, cadF, cdtA, cdtB, cdtC, ceuE*
36	*C. coli*	*cmeB, Ery75, Ery74, tet(O), tet(A), C257T*	*FlaA, cadF, cdtA, cdtB, cdtC*
37	*C. coli*	*cmeB, ermB, Ery75, tet(O), C257T*	*flaA, cadF, cdtA, cdtB, cdtC*
38	*C. coli*	*cmeB, ermB, Ery75, tet(O), tet(A) C257T*	*flaA, cadF, cdtA, cdtB, cdtC, ceuE*
39	*C. coli*	*cmeB, ermB, tet(O), tet(A), C257T*	*flaA, cadF, cdtA, cdtB, cdtC*
40	*C. coli*	*cmeB, ermB, Ery75, tet(O), tet(A), tet(B), C257T*	*flaA, cadF, cdtA, cdtB, cdtC, virB11*
41	*C. coli*	*cmeB, ermB, Ery75, tet(O), tet(A), C257T*	*flaA, cadF, cdtA, cdtB, cdtC, ceuE*
42	*C. coli*	*cmeB, Ery75, tet(O), tet(A), C257T*	*flaA, cadF, cdtA, cdtB, cdtC, virB11*
43	*C. coli*	*cmeB, Ery75, tet(O), tet(A), C257T*	*flaA, cadF, cdtA, cdtB, cdtC*
44	*C. coli*	*cmeB, ermB, Ery75, tet(O), tet(A), C257T*	*flaA, cadF, cdtA, cdtB, cdtC*
45	*C. coli*	*cmeB, ermB, tet(O), tet(A), tet(B), C257T*	*flaA, cadF, cdtA, cdtB, cdtC*
46	*C. coli*	*cmeB, Ery75, Ery74, tet(O), tet(A), tet(B)*	*flaA, cadF, cdtA, cdtB, cdtC*
47	*C. coli*	*cmeB, Ery75, tet(O), tet(A), bla_OXA-61_*	*flaA, cadF, cdtA, cdtB, cdtC, ceuE*
48	*C. coli*	*cmeB, ermB, tet(O), tet(A)*	*flaA, cadF, cdtA, cdtB, cdtC*

Ery75 = A2075G mutation; Ery74 = A2074C mutation.

## Data Availability

All data generated for this study are contained within this article/Appendix A.

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
