# Peer review of "Virulence Profiling, Multidrug Resistance and Molecular Mechanisms of Campylobacter Strains from Chicken Carcasses in Tunisia"

_antibiotics, 2022, doi:10.3390/antibiotics11070830_

Round 1
Reviewer 1 Report
Well done work.
Author Response
Dear Reviewer,
We are grateful for your positive evaluation of our manuscript. We have carefully checked the manuscript and hope this version will be acceptable for publication.
Thank you again.
Best regards
Awatef Béjaoui
Reviewer 2 Report
The article is written correctly and in detail. The conducted experiments are correctly performed and described. The subject taken up by the authors is extremely important due to the emergence of an increasing number of multi-drug Campylobacter strains. So is important to control the emergence and spread of such strains. However, the article does not introduce anything innovative apart from epidemiological data. Basic research that can be found in other articles is carried out. Moreover, the authors published similar studies in 2018. Hence, I have doubts about the interest in the article, and thus about the citation of the article.
In the article I found some issue which should be improved:
Line 105: Table 2, there are some spelling mistakes
Line 107: in the footer, abbreviations should be arranged according to their occurrence in the table
Abbreviations for genes coding for resistance or virulence factors should be explained in the description in the results and not just in the methods.
Why was sequencing not used in the identification of strains?
Line 377: the sequence number to which the tested sequences were compared should be given
Author Response
Dear Reviewer,
We thank you for your positive comments on our manuscript. We have taken into account all your comments and revised the manuscript accordingly.
Please find attached our point-by-point response to your comments.
Thank you again.
Best regards,
Awatef Béjaoui

Reviewer 3 Report
The manuscript is well written and uses appropriate methods for susceptibility testing and interpretation, although I would prefer MIC results so that trends can be more easily monitored over time for future research. The level of antibiotic resistance is quite high, especially concerning ciprofloxacin, a widely used empirical choice for treating GIT infections in humans. I would have liked to have seen data on the actual antimicrobial use at poultry farms in Tunisia. It is briefly commented on in the discussion that is is difficult to monitor since drugs are widely available, but should be included in the introduction in order to put the high levels of resistance into perspective for reader.
There are a few cases of plural disagreement, typos in the introduction:
Ln 67 : countries
Ln 71 - pathogen (no s)
Ln 72 - factor (no s)
Ln 73 - established (not ment)
ALSO, please be certain to define genes the first time used in the manuscript like you did for virB11, but not done for ceuE, cgtB or wlaN perhaps others.
Author Response
Dear Reviewer,
We are thankful for your positive comments on our manuscript and your suggestions. We have revised the manuscript taking into account all your comments.
Please find attached our point-by-point response, and we hope this version can be up to your requirements.
Thank you again.
Best regards,
Awatef Béjaoui
